

# The chemotactic swimming behavior of bird schistosome miracidia in the presence of compatible and incompatible snail hosts

Anna Marszewska[1], Anna Cichy[1], Jana Bulantová[2], Petr Horák[2] and Elżbieta Żbikowska[1]

[1] Department of Invertebrate Zoology and Parasitology, Faculty of Biological and Veterinary Sciences, Nicolaus Copernicus University in Torun, Torun, Poland
[2] Department of Parasitology, Faculty of Science, Charles University, Prague, Czechia

## ABSTRACT

No effective method has yet been developed to prevent the threat posed by the emerging disease—cercarial dermatitis (swimmer's itch), caused by infective cercariae of bird schistosomes (Digenea: Schistosomatidae). In our previous studies, the New Zealand mud snail—*Potamopyrgus antipodarum* (Gray, 1853; Gastropoda, Tateidae)—was used as a barrier between the miracidia of *Trichobilharzia regenti* and the target snails *Radix balthica*. Since the presence of non-indigenous snails reduced the parasite prevalence under laboratory conditions, we posed three new research questions: (1) Do bird schistosomes show totally perfect efficacy for chemotactic swimming behavior? (2) Do the larvae respond to substances emitted by incompatible snail species? (3) Do the excretory-secretory products of incompatible snail species interfere with the search for a compatible snail host? The experiments were carried out in choice-chambers for the miracidia of *T. regenti* and *T. szidati*. The arms of the chambers, depending on the variant, were filled with water conditioned by *P. antipodarum*, water conditioned by lymnaeid hosts, and dechlorinated tap water. Miracidia of both bird schistosome species chose more frequently the water conditioned by snails—including the water conditioned by the incompatible lymnaeid host and the alien species, *P. antipodarum*. However, species-specific differences were noticed in the behavior of miracidia. *T. regenti* remained more often inside the base arm rather than in the arm filled with water conditioned by *P. antipodarum* or the control arm. *T. szidati*, however, usually left the base arm and moved to the arm filled with water conditioned by *P. antipodarum*. In conclusion, the non-host snail excretory-secretory products may interfere with the snail host-finding behavior of bird schistosome miracidia and therefore they may reduce the risk of swimmer's itch.

## INTRODUCTION

Cercarial dermatitis (swimmer's itch) caused by bird schistosomes (Digenea: Schistosomatidae) is considered an emerging disease in Europe (*Horák et al., 2015*; *Tracz et al., 2019*). These parasites have a complex life cycle. The intermediate hosts are

Corresponding author
Anna Marszewska,
anna.marszewska@umk.pl

freshwater snails, whereas the final hosts are predominantly waterfowl. Snails infected with bird schistosomes release huge numbers of infective cercariae (*Soldánová, Selbach & Sures, 2016*). Thus, even with a low prevalence of bird schistosomes in intermediate host populations, there is a real threat for final hosts (*Marszewska et al., 2016*). As a result of the similarity of some lipid components in the integument of humans and birds, bird schistosome cercariae can accidentally penetrate the skin of people swimming or wading in the water (*Haas & Van de Roemer, 1998*). An attack by many infective larvae may be accompanied by some additional reactions, such as diarrhea, nausea, limb and lymph node swelling, and/or fever (*Horák et al., 2015*). The removal of freshwater snails, the source of infective cercariae, and waterfowl, the source of miracidia infective for the snails, from the environment to reduce the risk of human swimmer's itch provides equivocal effects (*Lévesque et al., 2002*; *Jouet et al., 2008*). Also, chemical methods such as the use of molluscicides may have an adverse impact on local fauna (*McCullough, 1992*). An increasing number of cercarial dermatitis outbreaks from year to year (*Marszewska et al., 2016*; *Tracz et al., 2019*), as well as alarming data on bird schistosome migration inside the body of experimental mammals (*Horák & Kolářová, 2001*; *Horák et al., 2008*) stimulated our interest in the biological control that would be applicable, especially in recreational water bodies.

Prevalence of digeneans in snails may be significantly reduced when a non-host snail species occurs in sympatry with the target snails and miracidia—the first larval stages of the parasites (*Kalbe, Haberl & Haas, 1997*). In detail, snail-finding by miracidia is influenced by several factors. The larvae react to, for example, the chemical cues secreted by snails (*Hertel et al., 2006*; *Seppälä & Leicht, 2015*), and light stimuli (*Gryseels et al., 2006*) with changes in movement. The larvae have a limited lifespan (*Anderson et al., 1982*) and may exhaust their penetration enzymes and energy reserves during unsuccessful penetration attempts (*Combes & Moné, 1987*), or they can penetrate into incompatible snails whose plasma kills the unspecific species of digenean trematodes (*Sapp & Loker, 2000*).

Free-living larvae of parasites and their hosts live in the midst of a complex biocenosis; as a result, their transmission takes place within a diverse community of non-host organisms that can have a huge influence on the success or failure of parasite transmission (*Hopper, Poulin & Thieltges, 2008*). The phenomenon of the reduction of disease risk as a result of the increased diversity of co-occurring non-host species is commonly known as the "dilution effect" (*Keesing, Holt & Ostfeld, 2006*; *Kopp & Jokela, 2007*; *Johnson & Thieltges, 2010*; *Cichy et al., 2016*). Studies have shown several ways how these non-host organisms can disrupt parasite transmission, for example, by acting as physical barriers (*Christensen, 1979*), predation, that is, active feeding on larvae (*Thieltges et al., 2008*; *Thieltges, Jensen & Poulin, 2008*; *Vielma et al., 2019*), non-host filter-feeders (*Mouritsen & Poulin, 2003*; *Hopper, Poulin & Thieltges, 2008*; *Marszewska & Cichy, 2015*; *Selbach, Rosenkranz & Poulin, 2019*) or decoys attracting the parasites (*Thieltges, Jensen & Poulin, 2008*). However, not all non-host species are of equal importance as diluters (*Hopper, Poulin & Thieltges, 2008*). For example, *Hopper, Poulin & Thieltges (2008)* demonstrated that only selected filter-feeding species have a significant influence on the transmission

of cercariae by feeding on these larvae, while other sympatric filter-feeders have no measurable effects on metacercariae formation in target second intermediate host. *Kopp & Jokela (2007)* showed that particularly alien species can be involved in the "dilution effect", that is, the snail species incompatible for an infectious agent can serve as a local protective shield for compatible hosts. For example, *Lymnaea stagnalis* (Linnaeus, 1758) (Gastropoda: Pulmonata), a species non-native to New Zealand, serves as a defender of the native *Potamopyrgus antipodarum* (Gray, 1843) (Gastropoda: Tateidae) against the trematode *Microphallus* sp. (Digenea: Microphallidae) (*Kopp & Jokela, 2007*). In our previous study (*Marszewska et al., 2018*), we made a successful attempt to experimentally use *P. antipodarum*, a snail species non-native to Europe, to interfere with the finding of a compatible lymnaeid host by bird schistosome miracidia.

On the other hand, it should be emphasized that nonindigenous animal species may become competent hosts for indigenous parasites (*Kelly et al., 2009*), or can be the source of pathogens with a possible/unpredictable effect on domestic fauna. In the native range, *P. antipodarum* is widely used by numerous digenean species, but there are no reports of bird schistosome infestation (*Hechinger, 2012*; *Selbach, Rosenkranz & Poulin, 2019*). However, *P. antipodarum* living in European waters is extremely rarely infected with a pre-patent and/or a patent invasion of digenean parasites (*Gérard & Le Lannic, 2003*; *Żbikowski & Żbikowska, 2009*; *Gérard et al., 2017*). In addition, *P. antipodarum* infection by some bird schistosome species failed under experimental conditions (*Rind, 1989*; *Marszewska et al., 2018*).

We hypothesize that any interference with snail host-finding by bird schistosome miracidia might influence the risk of swimmer's itch. Thus, the aim of the present research was to examine (1) whether the miracidia of bird schistosomes show totally perfect efficacy for chemotactic swimming behavior, (2) whether the larvae respond to substances emitted by incompatible snail species, (3) whether the excretory-secretory products of incompatible snail species disturb the search for a compatible snail host.

## MATERIALS AND METHODS

### Obtaining miracidia

Two specimens of definitive hosts—*Anas platyrhynchos* f. dom.—were experimentally infected by *Trichobilharzia regenti* (Digenea: Schistosomatidae), and two specimens of the same duck species were experimentally infected by *T. szidati* (Digenea: Schistosomatidae) according to the procedure described by *Meuleman, Huyer & Mooij (1983)*.

In order to obtain *T. regenti* eggs, the host nasal conchae were isolated during necropsy 20 days post infection (dpi), while the bird droppings were collected 18 dpi to obtain *T. szidati* eggs. Experimental duck hosts were euthanized by inhalation of isoflurane followed by decapitation.

The collected biological material with eggs was placed in dark long-necked flasks with dechlorinated tap water at 20 °C. After a few minutes, having hatched the first larval stages (miracidia) were clustered under the illuminated water surface. The miracidia were then individually collected with a micropipette.
## Animal ethics statement

Care and maintenance of experimental animals were carried out in accordance with European Directive 2010/63/EU and Czech law (246/1992 and 359/2012) for biomedical research involving animals. Experiments were performed with the legal consent of the Expert Committee of the Section of Biology, Faculty of Science, Charles University, Prague, Czech Republic, and the Ministry of Education, Youth and Sports of the Czechia under ref. no. MSMT-33740/2017-2

## Water conditioning process

As to the snail species, we prepared 3 types of conditioned water: (1) young individuals of *Radix lagotis* (Gastropoda: Pulmonata) (compatible host to *T. regenti*)—shell height of 6 (SE 0.1) mm; (2) young individuals of *L. stagnalis* (compatible host to *T. szidati*)—shell height of 9 (SE 0.3) mm; (3) adult parthenogenetic females of *P. antipodarum*—shell height of 3 (SE 0.2) mm. We used younger (smaller) lymnaeid snail individuals to increase the number of snails for water conditioning to reduce the impact of individual characteristics of particular specimens. Lymnaeid snails came from our laboratory breeding, whereas *P. antipodarum* came from the natural environment—Sosno Lake, Poland (53°20′15″N, 19°20′55″E); before the experiments they were examined by non-invasive method for the presence of patent invasions (*Żbikowska et al., 2006*). So far, no pre-patent invasion of *P. antipodarum* has been reported in Sosno Lake (*Cichy et al., 2017*; *Marszewska et al., 2018*).

The water conditioning process was conducted in beakers at room temperature. Inside each beaker, 5 g of living snails (97 individuals of *R. lagotis*, 60 individuals of *L. stagnalis* or 116 individuals of *P. antipodarum*) in 100 ml of dechlorinated water were kept for 2 h. This concentration was used to assure enough quantity of an emitted substance in case it had a low effect. After the incubation, the snail-conditioned water was filtered through filter paper. Only freshly prepared substances no older than 24 h were used in the experiment.

## Bird schistosome miracidial chemotactic swimming behavior

The experiment was carried out in choice-chambers (Fig. 1) designed by *Haberl et al. (1995)*. The larvae were released from the base arm (Fig. 1) with 1 ml of dechlorinated tap water to choose between two side arms. Depending on the variant (Table 1), the side arms were filled with 1 ml of water conditioned by *P. antipodarum*, water conditioned by lymnaeid hosts and/or dechlorinated tap water. Each variant of the experiment was performed in three series of 10 repetitions at 20 °C under standardized artificial even lighting. These series were conducted within a few minutes directly one after another.

In total, 30 replicates were performed for each parasite species per variant. One larva was placed in the base arm of the choice-chamber (Fig. 1) for each repetition. A new larva was used for each replicate. After 3 min, the junction between all the arms was closed and the location of the larvae was checked under a stereoscopic microscope. The entire arm including part of the migratory channel (Fig. 1) was considered the choice of the larva. The choice-chambers were thoroughly washed and dried before each use.

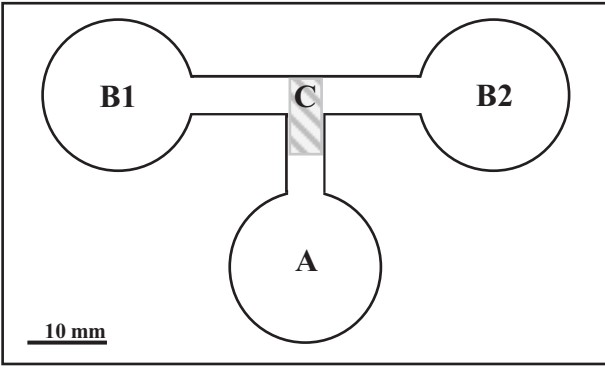

**Figure 1 The choice-chamber for evaluation of bird schistosome miracidial chemotactic swimming behavior (channel depth: 3 mm).** A, base arm; B1 and B2, side arms; C, closure.

**Table 1 The experimental variables.**

| No | First arm | Second arm |
|---|---|---|
| I* | Dechlorinated tap water | Dechlorinated tap water |
| II | Compatible lymnaeid host** | Dechlorinated tap water |
| III | Incompatible lymnaeid host*** | Dechlorinated tap water |
| IV | *Potamopyrgus antipodarum* | Dechlorinated tap water |
| V | Compatible lymnaeid host | Incompatible lymnaeid host |
| VI | Compatible lymnaeid host | *Potamopyrgus antipodarum* |
| VII | Incompatible lymnaeid host | *Potamopyrgus antipodarum* |

Notes:
* Control experiment.
** *R. lagotis* for *T. regenti* and *L. stagnalis* for *T. szidati*.
*** *L. stagnalis* for *T. regenti* and *R. lagotis* for *T. szidati*.

## Statistical analysis

We used a binomial test (*Zar, 1984*) to assess if the occupation of the side arms by larvae departed from the random distribution (assuming 50% probability of entering each arm) (Table 1). The tests were performed separately for each variant and species of the parasite, taking only the larvae present in the side arms (B1 and B2, Fig. 1) into account. Analyses were conducted in Microsoft Excel (version 2013).

## RESULTS

In the control experiment (dechlorinated tap water vs dechlorinated tap water) the miracidia of both bird schistosomes, *T. regenti* and *T. szidati*, occupied arms of the choice-chamber in similar proportions (Tables S1 and S2), and there were no significant statistical differences (a binomial test: $p = 0.4$, $p = 0.5$, respectively) (Fig. 2, I).

In the experiments with lymnaeid snail-conditioned water vs dechlorinated tap water (variants: II, III (Table 1)) statistically significant differences were observed in the distribution of miracidia (a binomial test: *T. regenti*—variant II: $p < 0.001$, variant III: $p < 0.01$; *T. szidati*–variant II: $p < 0.001$, variant III: $p = 0.001$) (Fig. 2, II–III). In variant IV, a higher number of larvae moved to *P. antipodarum*-conditioned water than to

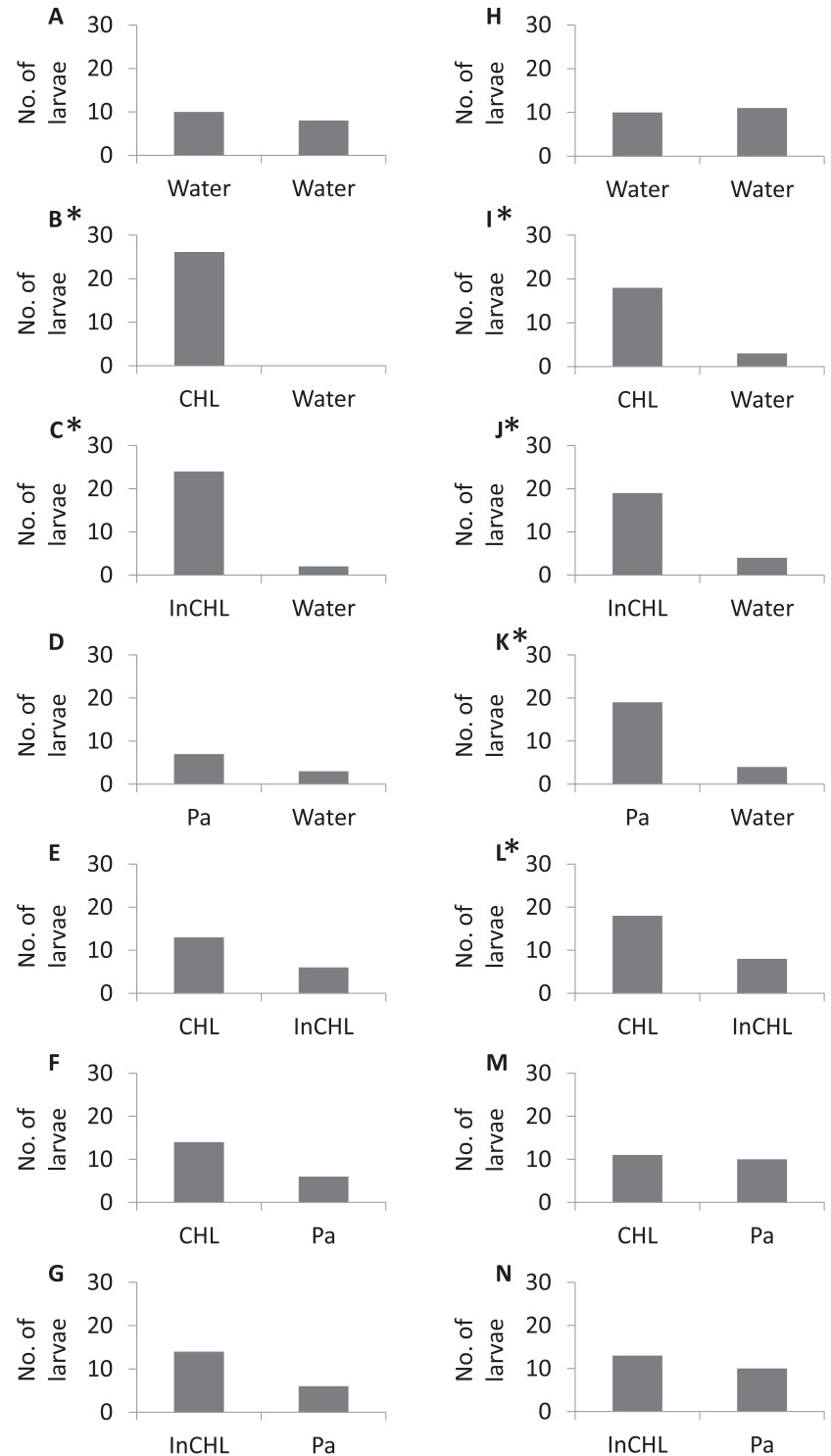

**Figure 2 The number of miracidia in the side arms per serie.** (A–G) Variants numbers from I to VII, respectively, for *T. regenti*; (H–N) variants numbers from I to VII, respectively, for *T. szidati*; Water, dechlorinated tap water; CLH, compatible lymnaeid host-conditioned water: *R. lagotis* for *T. regenti* and *L. stagnalis* for *T. szidati*; InCLH, incompatible lymnaeid host-conditioned water: *L. stagnalis* for *T. regenti* and *R. lagotis* for *T. szidati*; Pa, *P. antipodarum*; *Statistically significant a binomial test ($p < 0.05$).

dechlorinated tap water for both parasite species, but statistically significant differences were recorded only for *T. szidati* (a binomial test: $p = 0.001$) (*T. regenti*—a binomial test: $p = 0.2$). For the variant with *T. regenti*, only one-third of the larvae left the base arm. We were also able to observe statistically significant differences in variant V for *T. szidati* (a binomial test: $p = 0.04$). There was no significant statistical difference in variant V for *T. regenti* (a binomial test: $p = 0.08$). For both species of miracidia in variant V, we observed a higher number of the larvae in the arms filled by compatible lymnaeid host-conditioned water than those filled by incompatible lymnaeid host-conditioned water (Fig. 2, V).

There were no statistically significant differences in the variants VI and VII (lymnaeid snail-conditioned water vs *P. antipodarum*-conditioned water) (a binomial test: *T. regenti*—variant VI: $p = 0.06$, variant VII: $p = 0.06$; *T. szidati*—variant VI: $p = 0.5$, variant VII: $p = 0.3$). But, for miracidia of *T. regenti*, we observed a higher number of larvae in the arms filled by lymnaeid snail-conditioned water (compatible and incompatible) than by *P. antipodarum*-conditioned water, whereas for miracidia of *T. szidati*, the distribution of larvae in the arms was similar (Fig. 2, VI–VII).

## DISCUSSION

The majority of digenean species show a high specificity for their first intermediate hosts (*Sapp & Loker, 2000*). The phenomenon is also observed for species belonging to the genus of *Trichobilharzia* (*Kock, 2001*). It is also well known that many digeneans use chemo-orientation to find suitable hosts in the water column (*Haas, 2003*). *Kalbe, Haberl & Haas (1997)* showed that the miracidia preferred water conditioned by their specific host snails compared to incompatible snail species, and did not respond to water conditioned by leeches, tadpoles, and fish. Our study supports these findings and shows that bird schistosome miracidia significantly more often choose water conditioned by host snails than unconditioned water (Fig. 2, II). However, our results indicate that bird schistosomes respond also to the excretory-secretory products of non-host snails (Fig. 2, III), leading to the interference with miracidial chemo-orientation towards the host and non-host snail-conditioned water (*T. regenti*—Fig. 2, V, VI; *T. szidati*—Fig. 2, VI). These observations coincide with the results of many authors (*Kalbe, Haberl & Haas, 1996*, *2000*; *Haberl et al., 2000*; *Haas, 2003*; *Hassan et al., 2003*; *Kalbe et al., 2004*), who showed some strains of *Schistosoma mansoni* could not distinguish between their specific and five incompatible host species. This behavior of miracidia indicates the adaptive plasticity of the parasite or the similarity of individual components present in the range of snail-derived stimulators.

Generally, our results present differences in the accuracy of chemotactic swimming behavior of the miracidia of bird schistosomes. The miracidia of *T. regenti* react to substrates from the host snails and from the snails closely related to the suitable hosts to such an extent that it interferes with finding the substrate from the target species. The larvae of *T. szidati*, despite reacting to non-host lymnaeid snails, in the vast majority cope with finding a suitable substrate in the variant consisting of non-host lymnaeid snail-conditioned water and host lymnaeid snail-conditioned water. It is well known

that the used bird schistosome species migrate through the body of the final host in different ways, and also their eggs enter the external environment in other ways (*Horák et al., 2015*). Perhaps these differences then have their reflection in the accuracy of the first larval stages (hatched from the eggs) in search of a suitable host snail.

As to the species-specific reactions, *T. szidati* larvae more often choose water conditioned by *P. antipodarum* than dechlorinated tap water, while *T. regenti* larvae more often remain in the base arm (Fig. 2, IV). We know that the individual species of miracidia are adapted to the behavioral patterns of compatible snail species (*Behrens & Nollen, 1992*). *Lymnaea stagnalis* lives primarily at the water surface on aquatic plants, while young individuals of *Radix* spp. often form high-density populations covering the lake bottom in the shallow littoral zone, similar to *P. antipodarum* (*Piechocki & Wawrzyniak-Wydrowska, 2016*). We can only assume that the differences in the response of bird schistosome larvae to the *P. antipodarum*-conditioned water may be related to the above-mentioned difference between lymnaeid snail species.

Our observation is the first indication of these subtle differences in miracidial chemotactic swimming behavior, and the phenomenon merits further research.

## CONCLUSIONS

Based on our previous (*Marszewska et al., 2018*) and present results it seems that the increased biodiversity of malacofauna may interfere with the life cycle of bird schistosomes (snail host-finding and penetration), and thus it can potentially reduce the risk of swimmer's itch. To prove this possibility, we plan to test in the future not only the chemotactic behavior/chemical cues, but also the ability of bird schistosome miracidia to penetrate the incompatible snail host (*P. antipodarum* in our case). Moreover, currently conducted large-scale field research should answer whether this non-native snail is occurring in lakes at sufficiently high densities to interfere with the spread of swimmer's itch.

## ACKNOWLEDGEMENTS

We would like to thank Mrs. Hazel Pearson—a native speaker of English—for proofreading of an English version of the manuscript and Professor Jarosław Kobak (Nicolaus Copernicus University in Torun) for help in conducting statistical analysis.

### Funding

This work was supported by the grant of the National Science Centre, Poland No. 2017/25/N/NZ8/01345. The funders had no role in study design, data collection and analysis, decision to publish, or preparation of the manuscript.

### Grant Disclosures

The following grant information was disclosed by the authors:
National Science Centre, Poland: 2017/25/N/NZ8/01345.
## Competing Interests

The authors declare that they have no competing interests.

## Author Contributions

- Anna Marszewska conceived and designed the experiments, performed the experiments, analyzed the data, prepared figures and/or tables, authored or reviewed drafts of the paper, and approved the final draft.
- Anna Cichy performed the experiments, authored or reviewed drafts of the paper, and approved the final draft.
- Jana Bulantová performed the experiments, authored or reviewed drafts of the paper, and approved the final draft.
- Petr Horák conceived and designed the experiments, authored or reviewed drafts of the paper, and approved the final draft.
- Elżbieta Żbikowska conceived and designed the experiments, authored or reviewed drafts of the paper, and approved the final draft.

## Animal Ethics

The following information was supplied relating to ethical approvals (i.e., approving body and any reference numbers):

Experiments have been performed under the legal consent of the Expert Committee of the Section of Biology, Faculty of Science, Charles University, Prague, Czech Republic, and the Ministry of Education, Youth and Sports of the Czech Republic under ref. no. MSMT-33740/2017-2.

## Data Availability

The raw data are available in the Supplemental Files.

## Supplemental Information

Supplemental information for this article can be found online at http://dx.doi.org/10.7717/peerj.9487#supplemental-information.

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
