# Peer review of "The chemotactic swimming behavior of bird schistosome miracidia in the presence of compatible and incompatible snail hosts"

_PeerJ, doi:10.7717/peerj.9487_

## Round 0.1 · original submission · Major Revisions

Dear authors,

Thank you for submitting your manuscript to PeerJ. It has been reviewed by three experts in the field who have provided very helpful comments for improvement. All referees consider that the study is interesting but also made substantial criticisms, particularly in the experimental design and statistical analysis. There are also a number of concerns on the choice chamber that should be clarified. Referee three also made important comments especially in the introduction and discussion section that should be considered. I ask you to pay particular attention to these aspects and invite you to carefully revise your paper in light of all the comments provided.

Yours sincerely

Reviewer 1 ·

Basic reporting

The English is generally fine, but some passages need rephrasing or polishing. For example, Table 1, “Variants of filling the side arms” is extremely awkward writing. And so is “The place of closing the arms” in the caption of Figure 1.

The Introduction is narrowly focused on digeneans, specifically schistosomes. There is an extensive literature on transmission interference by non-compatible hosts acting as decoys, sinks, etc (see Thieltges et al. 2008 Parasitology). It would be nice to have a broader context presented in the introduction.

Experimental design

The snail Potamopyrgus antipodarum is exploited by a rich fauna of digeneans in its native New Zealand. It would be important for the authors to tell us about the parasites of this snail. Are any of those schistosomes? Could it be that this snail is already compatible to other schistosomes?

Validity of the findings

The results of all statistical tests should be presented by including the test statistic, and not just the P-value.

Also, I don’t think Kruskal-Wallis tests are appropriate. Because the output data are categorical with three possible outcomes (miracidium chooses one arm, the other arm, or neither), the authors should use a multinomial GLM, using each replicate trial as an observation, with the three series as random factor.

The possibility that Potamopyrgus antipodarum interferes with miracidial host-finding and reduces the risk of swimmer’s itch depends entirely on its abundance and distribution. The authors provide no data on this. Is this non-native snail occurring in lakes with swimmer’s itch at sufficiently high densities to cause interference? If not, then the conclusions are a little exaggerated.

Additional comments

This is a simple but well-designed experiment testing the attraction of miracidia of two species of schistosomes to the odours of their snail host versus the odours of non-compatible snails. I feel it could be accepted, but first it requires some revision.

Reviewer 2 ·

Basic reporting

1. There are some minor typographical and grammar errors including:
a. Line 191 – the space is missing between “regenti” and “larvae”.
b. The caption for Figure 1 is missing a closing parenthesis.
c. In Lines 20, and 80-81, the authors have written “incompatible snails species”, the plural of snails is superfluous.
d. Line 27 – A sentence should not start with an abbreviation
e. Line 96 – Suggest using the appropriate term for an autopsy performed on an animal.
f. Lines 143/149 – Suggest removing the first-person phrasing.
g. Line 78 – Suggest replacing “check” with a more appropriate word.

Experimental design

1. There is some additional information required for another researcher to be able to successfully replicate this study.

a) i) Did the larva have to be fully in the choice chamber to be counted as such? What about larvae that remained in the “C” area or the arms leading to the “B” areas of the choice chambers? What distance from the choice chambers was chosen for the larva to be considered to have preferred that water treatment?
ii) In the supplemental tables the authors provided, all of the series values for each variant add up to 10, does this mean that in every single replicate the larva ended up entirely in one of the arms?
b) Were the choice chambers cleaned thoroughly between each replication? If not, it is possible that a higher concentration of snail excretory products was present than intended.
c) The authors state that one larva was used for each repetition, does this mean that each larva was used only once and replaced, or that one larva was consistently used in all 10 repetitions in a series?

2. The sample size for this study is adequate only if a new larva was used in each repetition (n=60), otherwise with pseudoreplication, n=6 (3 series for each parasite species) which is not sufficient.

3. In the statistical analysis section of the methods, the authors state that they used the sum of the larvae in the arms from each of the variants to get a value for every arm in each of the six series. But, in Figure 2, they report the average number of miracidia in the arms per series. It is misleading, because when the authors report the results, they give the p-value from the statistical analyses and cite Figure 2, but those should be two separate entities. These need to match.

4. In the results section, the authors report the p-values for the variants, after which they have a statement regarding the percentage of larvae found in the specific choice chamber for that variant. If the percentages are mentioned, they should be included as a numerical value. It is not necessary to mention the percentages when the results are insignificant (i.e. variants V, VI, VII).

Validity of the findings

1. A possible confounding factor in this study is that the Potamopygyrus antipodarum snails may have had pre-patent infections. Is it possible that a snail with a pre-patent infection excretes a different chemical cue than a snail that is free of infection?
2. Why were young individuals of the lymnaied snails and adults of P. antipodarum used in this study? Since the lymnaeid snails were taken from lab colonies, why not keep it consistent across all snail species and use adult lymnaeids?

Additional comments

This paper discusses using an incompatible host species of snail to prevent miracidia from finding their compatible host species as a method for controlling swimmer’s itch. The research gap identified in the study was the use of miracidial behaviors and preferences as a means of reducing the risk of swimmer’s itch instead of killing hosts or releasing a chemical into the environment. This was evident in the Introduction and Conclusions but was lacking in the Discussion section. The research outlined in this paper is within the scope of this journal and the authors have well-defined research questions. The proper controls were in place, the required ethical information was present, and the experiments were carried out ethically. There is some information missing from the Materials and Methods section that would be required to replicate this experiment, which is addressed below. The Conclusions section linked back to the original research questions and acknowledged that there were still further studies that were required to answer those questions.

The authors found that miracidia of both Trichobilharzia szidati and T. regenti preferred the lymnaeid snail-conditioned water over the dechlorinated tap water, whether it was from the compatible or incompatible host. And, when given the choice between the compatible or incompatible lymnaeid snail-conditioned water, there was no significant difference between the two chambers for both species. As for the tests with the non-native snail-conditioned water, T. regenti appeared to avoid it, while T. szidati chose the snail-conditioned water more than the dechlorinated tap water. When given the choice between the non-native snail-conditioned water and lymnaeid host-conditioned water, compatible or incompatible, there were no significant differences between the choice chambers for the miracidia of both species.

In the introduction, the authors clearly explained their reasoning as to why the reader should be concerned about swimmer’s itch, and why it is important to find a way to combat it without negatively impacting the local fauna. The authors clearly articulate why the incompatible snail species may be able to act as a shield for compatible snail hosts. However, there are a few ways to improve the introduction, including explaining what the miracidia and cercariae larval stages are with respect to the schistosome life cycle. This will help the reader to better understand the authors’ points in the first paragraph. Also, it would help to affirm the authors’ message if it was explained that snails can release thousands of cercariae during a trematode infection. Therefore, decreasing snail infections through the use of methods explained in this paper could greatly help in reducing the occurrence of swimmer’s itch. The authors touch on this point very briefly in Lines 77&78, but it should be expanded upon.

Aesthetically, the figures are clear and easy to understand. As for the content, there is some confusion on the results reported in Figure 2, which is addressed below. Table 1 does a really good job of explaining all of the variants used in the study and was the most concise way to describe them, as writing them out in the results section likely would have been cumbersome and confusing.

For the most part, the manuscript was well written and straightforward. There are some places where the phrasing could be improved, such as in Lines 27-29, 59, 62-63, 68-70, 71-74, and 125-126 to aid in reader comprehension.

·

Basic reporting

Please find my detailed remarks summarized in the general comments section below.

The English language is generally understandable but could benefit from some proof-reading and editing, e.g. in lines 62-63.

Raw data is provided.

Experimental design

The experimental setup is well-designed and appropriate to test the authors’ research question. Materials and methods are described in adequate detail to allow replicate studies (see some minor comments below though).

L110-113: It would be helpful to highlight which of the snail species represent compatible/incompatible hosts to the respective trematode miracidia.

Validity of the findings

See below.

Additional comments

The authors present a study on the chemotactic swimming behaviour of bird schistosome miracidia in the presence of compatible and incompatible snail host species. Since bird schistosomes are important infectious disease agents in recreational water bodies in Europe, this study is of relevance and should be of interest to the scientific community.

However, the manuscript has some shortcomings and weaknesses, in particular in the introduction and discussion sections, that would need to be addressed before considering the work for publication. Unfortunately, I have to recommend major revisions of the manuscript in its current from. I hope these comments can be useful for a successful re-submission of this undoubtedly relevant and interesting work.
* * *
In detail, I have the following main concerns and objections: Overall, the introduction and discussion sections seem a bit rushed and could be expended.

1. I believe it would be very helpful to include a brief overview of the general bird schistosome life cycle, as not all readers will be familiar with it. Furthermore, I suggest explaining the typical host-specificity of these parasites to their first intermediate snail host, since this is a central element of the study’s research question.

2. One of the key aims of this study was to test whether “miracidia of bird schistosomes exhibit chemotactic swimming behaviour” (L78-79). However, the authors already answer this question in the introduction (L58-59). Furthermore, this has also been investigated for bird schistosomes specifically and some of the relevant literature is already cited by the authors. The following reference could be added in this context: Hertel, J., Holweg, A., Haberl, B., Kalbe, M., and Haas, W. (2006). Snail odour-clouds: spreading and contribution to the transmission success of Trichobilharzia ocellata (Trematoda, Digenea) miracidia. Oecologia 147, 173–80.

3. I am critical of the interpretation that “as a result of the closer micro-habitat of suitable snail hosts (Radix spp.) and non-host snails (P. antipodarum), T. regenti probably developed a tool to avoid penetration of a non-host snail” (197-199).
Potamopyrgus is a rather recent invader in European and North American waterbodies, and likewise, lymnaeid snails were only recently introduced into New Zealand freshwaters, the native habitat of P. antipodarum. Accordingly, the host-parasite co-evolution of bird schistosomes and their snail hosts (or potential dead-end diluters) has occurred for the longest time without contact to P. antipodarum. In the case of Potamopyrgus/Trichobilharzia such an adaptation would therefore have occurred quite recently and very quickly, which would require strong selective pressure. I am absolutely in favour of speculation but I would suggest a more cautious interpretation of these results, if possible backed by and discussed in the context of findings from comparable studies on such evolutionary processes.
* * *
Furthermore, the following minor comments should also be considered by the authors:

• L1: The title seems a bit wordy to me. Suggestion for a slightly shorter version: “The chemotactic swimming behavior of bird schistosome miracidia in the presence of compatible and incompatible snail hosts”
• L13, and throughout the text: Please change “invasive” to “infective” when speaking of cercariae/miracidia. Potamopyrgus antipodarum is an invasive species.
• L16: Please change “application” to “presence”
• L57: Please change “between” to “in sympatry with”
• L72: “Marszewska et al., 2018b” is cited before “Marszewska et al., 2018a” (L208)
• L90-95: Move to separate ethics section.
• L106: Please add the number of snails used.
• L173: change “Digenea” to “digenean”
• L178: Suggestion “Our study supports these findings and shows…”
• L181 and following: Please remove “statistically significant” in the discussion.
• L191: add space between “regenti” and “larvae”
• L195: pleases change to “covering the lake bottom in the shallow littoral zone, similar to…”
• L196: Is “mimic” the correct term here? I would suggest “are adapted to”
• L215 following: Please check the references. Species names should be in italics; journal names should use capital letters.
• Figure 1: I would suggest labelling the side arms “B1/B2” to show that they contain different treatments.
* * *
Despite my critical points, I commend the authors on their work and I hope my suggestions can be useful for a successful re-submission of this study.

---

## Round 0.2 · Minor Revisions

Dear co-Authors

Thank you for your effort in improving the ms. It has been considerably ameliorated and I’m satisfied with most of the changes you made.

However, in addition to the improvement of English and minor changes suggested by referees, I’m still concerned by the statistical analysis. I consider that explanation of statistical analysis (currently explained in two sentences) is very poor and important information is required. For example, what kind of analysis did you conducted (GLM)? What was the link function (logit)? Which were the variables used in the model? How did you select the best model (Forward, backward, stepwise,…)? which statistical program did you use? Etc. It would be also important to include a table with the results of the models, including the values of estimates, etc.

Minor comments:
Lines 149-150 “This concentration was used to rule out the possibility that the emitted substances would have a low effect”. I imagine you want to say that this concentration was used to assure enough quantity of emitted substance in case it has a low effect.

Reviewer 1 ·

Basic reporting

The English writing remains generally poor in many places (grammar and spelling), and ok at best elsewhere. It needs improving either by a fluent English speaker familiar with science writing, or by a professional editing service.

Experimental design

My earlier concerns have been satisfactorily addressed.

Validity of the findings

My earlier concerns have been satisfactorily addressed.

Additional comments

My earlier concerns have been satisfactorily addressed, but the writing (in many places) still falls short of what I would consider acceptable.

·

Basic reporting

see below

Experimental design

see below

Validity of the findings

see below

Additional comments

I acknowledge the authors’ efforts to include all suggestions and feedback from the three reviewers in the revised version of their manuscript. I don’t have any further major comments and can suggest the manuscript for publication in PeerJ.

Congratulations to the nice study.

The authors might want to consider the following minor comments though:
- L52: Some more information on the disease could be added here, e.g., “bird schistosome cercariae can accidentally penetrate the skin of people swimming or wading in the water, where they can cause an allergic reaction and an itchy rash.”
- L60: please add a comma after “applicable”
- L74: Maybe “non-host species” is more suitable than “potential targets,” since the following examples include predators, filter feeders etc.
- L76-77: I feel the beginning of this sentence is a bit vague. Maybe something like “Studies have shown several ways how these non-host organisms can disrupt parasite transmission, e.g., by acting as physical barriers....” would work?
- L81: I would suggest adding “as diluters” after “importance”
- L94: Please change to “schistosome miracidia”
- L95-103: I feel this paragraph might be better placed in the discussion section.
- L148: “were kept in 100 ml of dechlorinated water for 2 h.”
- L203: “It is also well known that many digeneans use chemo-orientation to find suitable hosts in the water column.”
- L231: Full species name at the beginning of sentence should be used.

---

## Round 0.3 · Major Revisions

Dear co-authors,

Thank you for your revised version. However I have still concerns about the statistical analysis you performed and the reasons for not conducting a GLMM analysis. If you consider only two possible outcomes for your response variable (binary data) you can still perform a GLMM as suggested by the referee, but with a binomial distribution and a logit link function. By using AIC you can compare models using the three series of repetitions as random factor (GLMM), with models without the random effect (GLM), to know which is the better model explaining your data. If there is a reason why you consider such analyses are not appropriate for your data, please provide an explanation in the rebuttal letter. Thank you very much for your effort!

---

## Round 0.4 · accepted · Accept

Dear co-authors,

Thank you for your revised version. Regarding the analysis of your data, a GLM with a binomial error distribution would allow you to analyse the proportion of larvae that chose a given arm, with the species of parasite and the treatment as response variables. But if you confirm that you are not interested by these effects (and you removed the random effect), and you want just to know the individual choice, performing multiple test, you can keep the binomial test, although I think that you can get more out of your data!